# Valley Fever: Environmental Risk Factors and Exposure Pathways Deduced from Field Measurements in California

**DOI:** 10.3390/ijerph17155285

**Published:** 2020-07-22

**Authors:** Antje Lauer, Vicken Etyemezian, George Nikolich, Carl Kloock, Angel Franco Arzate, Fazalath Sadiq Batcha, Manpreet Kaur, Eduardo Garcia, Jasleen Mander, Alyce Kayes Passaglia

**Affiliations:** 1Department of Biology, California State University Bakersfield (CSUB), Bakersfield, CA 93311-1022, USA; ckloock@csub.edu (C.K.); afrancoarzate@csub.edu (A.F.A.); fazalathbatcha@gmail.com (F.S.B.); mkaur@student.roseman.edu (M.K.); egarcia61@csub.edu (E.G.); jmander@csub.edu (J.M.); alyce.passaglia@sjsu.edu (A.K.P.); 2Desert Research Institute (DRI), Las Vegas, NV 89119, USA; vic.etyemezian@dri.edu (V.E.); George.Nikolich@dri.edu (G.N.)

**Keywords:** *Coccidioides*, coccidioidomycosis, dust, exposure, fungus, health hazard, military, Mojave Desert, pathogen, PM10, soil, Valley fever

## Abstract

Coccidioidomycosis, also known as Valley fever, has been reported among military personnel in *Coccidioides*-endemic areas of the southwestern United States since World War II. In this study, the prevalence of *Coccidioides* was confirmed in different soil and dust samples collected near three military bases in California using DNA extraction and Polymerase Chain Reaction (PCR) methods. Analyses of physical and chemical parameters revealed no significant differences between *Coccidioides*-positive and -negative sites. Soil samples collected in the Mojave Desert (near Twentynine Palms MCAGCC) showed the highest percentage of *Coccidioides*-positive soil and dust samples. Samples from the San Joaquin Valley (near NAS Lemoore) showed the lowest percentage of positive samples and were restricted to remnants of semi-natural areas between agricultural fields. Our results suggest that soil disturbance around all three military bases investigated poses a potential *Coccidioides* exposure risk for military personnel and the public. We conclude that once lands have been severely disturbed from their original state, they become less suitable for *Coccidioides* growth. We propose a conceptual framework for understanding exposure where disturbance of soils that exhibit natural or remnants of native vegetation (Creosote and Salt Bush) generate a high risk of exposure to the pathogen, likely during dry periods. In contrast, *Coccidioides*-positive sites, when undisturbed, will not pose a high risk of exposure.

## 1. Introduction

### Coccidioides spp. and Coccidioidomycosis

Coccidioidomycosis, also known as San Joaquin Valley fever, is caused by two pathogenic fungi belonging to the genus *Coccidioides*, *Coccidioides immitis* and *Coccidioides posadasii*. These pathogens are obligate aerobic dimorphic fungi that form vegetative hyphae (mycelium) when growing as saprophytes in the top horizons of moist soil and are somehow adapted to a variety of environments that could be characterized as the Lower Sonoran Lifezone [1]. This zone comprises arid and semi-arid areas in the southwestern U.S. and habitats with similar climate conditions, in Mexico and South America [2,3,4,5,6,7,8,9,10,11]. During the dry season, *Coccidioides* spp. survive in the soil in the form of barrel-shaped arthroconidia which can easily become airborne when soil is disturbed. These arthroconidia are approximately 2–5 μm in length, which is small enough to remain suspended in the air for hours to days and to be inhaled into the lungs, where an infection may result [12]. It appears that the keratinophilic pathogen is most successful in soils that contain organic material from small mammals [13], and where few other microbial species compete for resources, as it has been termed a poor competitor which benefits from reduced antagonism by other microbes in alkaline-saline soils of the American deserts [9,14].

Although approximately 60% of infections are asymptomatic, and most other patients experience self-limited influenza like symptoms, rashes, and fatigue [15], some patients develop more severe symptoms that can lead to dissemination to skin, bone and meninges of the brain. At highest risk for disease complication and dissemination are immunocompromised patients including pregnant women, the very old and the very young, and patients suffering from diseases that affect the immune system, such as HIV [16]. Furthermore, humans of African or Filipino descent are considered at higher risk for disease dissemination, likely due to genetic predisposition [17].

Coccidioidomycosis is believed to be profoundly underdiagnosed and its impact on lost productivity grossly underappreciated [18]. To characterize long-term national trends, the Center of Disease Control and Prevention (CDC) regularly analyzes data from the National Notifiable Diseases Surveillance System (NNDSS) [19]. The incidence of reported annual coccidioidomycosis in the general public increased substantially between the late 1990s and 2018 in Arizona and California with peaks after the interruption of a long-term drought in 2004, 2010 and in 2017 (Appendix A; see also [20]). The development of a vaccine has been elusive to date, and accurate diagnosis and treatment of the disease can be difficult [18]. Therefore, prevention through reasonable reduction in exposure is likely the best way to reduce incidence of the disease, human suffering, and the financial burden on health care [21,22].

An excellent source of documentation for when and where humans may have been exposed to the pathogen in different environments in the southwestern U.S. are military bases [23,24,25,26,27,28,29], because most of the personnel is stationary. In addition to reported cases among military personnel on bases in endemic regions of the pathogen, coccidioidomycosis was also occasionally documented on bases in traditionally non-endemic areas (Figure 1). Several large military bases are situated in highly endemic areas of *Coccidioides* spp. in California and Arizona, such as Edwards Air Force Base (EAFB) and Marine Corps Air Ground Combat Center (MCAGCC) Twentynine Palms, both in the Mojave Desert in California, as well as Davis-Monthan Air Force Base (DMAF) near Tucson, Arizona. Cases of Valley fever have been frequently reported among military personnel training in *Coccidioides*-endemic areas since World War II [30,31,32,33,34,35,36,37]. Data accumulated by Reeves and Kugblenu (2016) [38] showed that 216 military personnel of the U.S Air Force have died from fungal infections between 1970 and 2012, 32 of them (14.81%) confirmed cases of coccidioidomycosis with a high number of deaths through unspecified mycoses (43%). Recently, disease incidence among the active component of the U.S. armed forces appear to have been highest at the Lemoore Naval Air Station (NAS), Edwards Air Force Base (AFB), China Lake Naval Air Weapon Station (NAWS), and the Fort Irwin National Training Center (NTC), all in California, as well as at Davis-Monthan Air Force Base (AFB) near Tucson and Luke Air Force Base (AFB) near Phoenix, both in Arizona (Figure 1A,B).

Santa Ana winds can carry dust from *Coccidioides*-endemic areas in the Mojave Desert west to the Los Angeles Basin and beyond, generating a potential health risk for people in non-endemic areas of the pathogen, including tourists but also marine mammals [39,40]. In addition, military training activities in arid areas reduce vegetation cover, disturb Biological Soil Crusts (BSCs) that are protecting soils from erosion by wind in desert environments [41,42]. The impacts of military vehicle traffic on natural areas were addressed by Anderson et al. (2005) [43] and by Svendsen et al. (2017) [44] and concerns were raised about off-road recreational activities in semi-arid areas of the southwestern U.S. Recovery attempts of soils and vegetation in military base and training camps were conducted after World War II in some areas of the Mojave and Sonoran Deserts [45,46,47]. These studies indicate that recovery of compromised desert soils and vegetation to pre-disturbance status takes at least several decades, sometimes more than a century.

Furthermore, the diminishing groundwater in the San Joaquin Valley and the Mojave Desert is putting additional strain on air quality and coccidioidomycosis incidence because the ongoing drought in recent years has led to an increase in fallow agricultural fields, allowing the pathogen to become re-established, generating additional dust exposure pathways in *Coccidioides*-endemic areas [48,49,50,51,52].

In addition to military personnel being affected by coccidioidomycosis, an abrupt increase in the rates of coccidioidomycosis has been documented for construction workers and farm laborers [53], children [54], and incarcerated populations [55,56] which contributed to increasing rates of hospitalizations for the disease [57]. There is widespread perception and modeling evidence that the southwestern U.S. is becoming not only drier, but also dustier [58,59]. The areas where a significant decrease in precipitation is predicted are also highly endemic for *Coccidioides* and the expansion of *Coccidioides* habitat due to climate warming has been predicted [60,61].

It has been hypothesized that exposure to infectious fungal arthroconidia is related to the emission of dust from arid soils where the pathogen is endemic, due to disturbance by humans or due to natural causes. This apparent link is often difficult to support with direct evidence, and often it is not clear if exposure is a result of regional high winds that suspend arthroconidia of the pathogen from large, diffuse source regions within the endemic areas (which can encompass numerous “hot spots”), or if there are a limited number of much smaller source areas that are the primary culprits for infection. More recently, the source of the pathogen was linked successfully to outbreaks of coccidioidomycosis especially among construction workers [62,63].

The overall goals for this exploratory study were therefore(i)To assess whether *Coccidioides* DNA can be detected in soils in study areas near military bases in California were coccidioidomycosis cases were reported in previous years;(ii)To investigate whether the presence of *Coccidioides* is supported in soils that are characterized by specific pH, soil texture, soil ionic content, and possibly other parameters (identified parameters could be linked to large-scale datasets, or maps to indicate the distribution of supportive *Coccidioides* habitats);(iii)To investigate whether wind-suspendable dust at sites where surface soils do test positive for the pathogen’s DNA, and to investigate whether dust suspended by travel on unpaved roads in endemic areas of *Coccidioides* are potential or significant pathways for exposure.

The overall, long-term intent is to develop a conceptual framework for how exposure to the pathogen occurs at specific sites where the fungus is known to grow regionally within endemic areas.

## 2. Material and Methods

Soil and dust samples were collected from surface soils near three military facilities located in Valley fever-endemic areas in California with prior history of disease incidences among military personnel at different times of the year. Those were Naval Air Station (NAS) Lemoore in the San Joaquin Valley, Edwards Air Force Base (AFB) in the western Mojave Desert, and Marine Corps Air Ground Combat Center (MCAGCC) Twentynine Palms in the southeastern Mojave Desert. In addition to bulk soil samples (5–7 cm depth in general), soil core samples were collected as well, mostly from sites where *Coccidioides* DNA was detected. Wind-suspendable dust was collected for analysis using a wind tunnel-like device from a subset of sites. Similarly, samples of resuspended dust from vehicle travel on unpaved roads were collected from a very limited number of road segments within the three study areas. Soil samples were analyzed for physical and chemical constituents using standard methods. Biological testing relied on DNA extraction and analysis. Detailed site descriptions, as well as field and lab method descriptions are provided below. Table 1 summarizes the number of soil and dust samples collected over the time of this project.

### 2.1. Site Selection

NAS Lemoore, Lemoore, CA: NAS Lemoore has ~20,000 active military, civilians and dependents and obtains approximately 6.4 inches of rainfall annually. Soil types in this area include sandy loams and fine sandy loams, various clay loams (some saline-sodic), and clay. The installation is in Kings County and Fresno County in the Central Valley of California (36°20′2″ N, 119°57′6″ W) and known as the Navy’s newest and largest Master Jet Base. The base itself is embedded within agricultural fields with very few sites remaining that show native vegetation or small mammal and other animal communities. Agricultural activities such as ploughing and harvesting result in regular dust emissions in the spring and fall seasons.

Eighty-two military beneficiaries at NAS Lemoore were diagnosed with coccidioidomycosis between January 2002 and December 2006; overall, a ~10-fold increase in incidence among active duty military personnel was observed in 2006 compared to 2002 [64]. Samples were collected from agricultural fields, orchards, meadows and dirt roads between fields. See Figure 2 for an overview of all sampling sites around NAS Lemoore.

Antelope Acres west of Edwards AFB, Rosamond, CA: Edwards AFB has ~18,000 active military, civilians and dependents. The area in Kern County, CA, where this military base is located (34°53′07″ N, 117°49′56″ W) receives approximately 7.4 inches of rainfall annually. Major soil types included Adelanto coarse sandy loams and Cajon loamy sand. The Mojave Desert shows much less human influence compared to the Central Valley of California where NAS Lemoore is located. Soil disturbance in the Mojave Desert often results from recreational off-road dirt biking and from minor agricultural activities around larger cities, such as Lancaster and Palmdale in the western Mojave. In contrast, in the San Joaquin Valley, where NAS Lemoore is located, the natural landscape has changed to industrial agriculture, resulting in only few remaining semi-pristine habitats [65]. Nevertheless, substantial swaths of area are severely eroded near NAS Lemoore, and are covered with invasive grasses or no vegetation with occasional brush. The western Mojave Desert is undergoing intense renewable energy constructions, especially solar power developments around the community of Antelope Acres. Close to the base and the dry lakebeds extensive Salt Bush (*Atriplex* spp.) areas and Creosote (*Larrea tridentata*) are common, and the area is known for spectacular native wildflower blooms in the spring. Incidence of coccidioidomycosis is high in this area. Windblown dust emissions generally occur during periods of high westerly wind, which are common and probably are what have led to an increase in coccidioidomycosis in humans over the last decade during times of increased soil disturbance [65,66]. Prior research in this area by our group has confirmed the presence of DNA of the pathogen in soil and dust [66]. See Figure 3 for an overview of all sampling sites in a highly disturbed area west of EAFB.

Twentynine Palms MCAGCC, CA: The MCAGCC, which is the largest United States Marine Corps, is situated in the southern Mojave Desert east of Yucca Valley in southern San Bernardino County (34°13′54″ N, 116°03′42″ W). It hosts ~50,000 active military, civilians and dependents. The area around the base is characterized by scattered desert shrub, mostly natural vegetation with *Larrea tridentata*, *Atriplex* spp., occasional cacti (commonly *Opuntia basilaris* and *Cylindropuntia ramosissima*), and Joshua trees (*Yucca brevifolia*), and receives approximately 4.4 inches of rainfall annually. Incidence of coccidioidomycosis is comparatively low in this rather undisturbed environment.

Sampling sites in the Mojave Desert near MCAGCC showed scattered vegetation with large areas of exposed soils, vulnerable to erosion when not covered with Biological Soil Crusts (BSCs). The presence of rodent communities, especially ground squirrels (*Otospermophilus beecheyi*), as well as large colonies of *Pogonomyrmex* spp. which contribute to soil disturbance were observed. No soil data were available from the United States Department of Agriculture (USDA) websoilsurvey (WSS) database for this area. However, data were available for sampling sites in nearby Joshua Tree National Park (NP). These soils were characterized as Pintobasin gravely sand, Werewolf gravely sandy loam and Morongo, Pinecity Cajon and Nasagold loamy sands (landforms: fan aprons and basin floors), typic Haplosalids (clayey), embedded in Jumborock outcrop associations (Figure 4).

### 2.2. Soil and Dust Sampling

Bulk soil samples were collected in January/February (winter), May/June (spring/summer), and September (fall) 2017 from all study areas. Samples (~40 g per samples) were collected into sterile 50 mL Falcon tubes by trowel to a depth of 5–7 cm depth. Near each facility where sampling occurred, an attempt was made to collect soil samples from locations that maximize variety among parameters, such as soil texture (grain size) and other physical and chemical properties (e.g., pH and total salts), vegetation cover, bioturbation (mixing action of soil by burrowing rodents and other animals). The first sampling event focused on obtaining a variety of soil samples to represent the variation in these parameters, whereas the second and third sampling events focused predominantly on those sites where *Coccidioides* DNA was detected, and only included re-sampling for a subset of the negative sites, due to budget constraints. However, this allowed us to re-sample and analyze all *Coccidioides*-positive sites in more detail than would have been possible if we had resampled all sites during all seasons. Consequently, this meant that a rigorous seasonal analysis of all sites could not be performed because not all *Coccidioides*-negative sites were re-visited during spring/summer and fall sampling. The spring/summer and fall bulk surface sample sets were supplemented with several core samples (surface to ~30 cm depth in ~5 cm intervals) that were collected mostly from sites that tested positive for the pathogen’s DNA in the winter (Edwards AFB: 7 cores from 5 sites, NAS Lemoore: 7 cores from 4 sites, Twentynine Palms MCAGCC: 14 cores from 10 sites).

Wind-suspendable dust samples were collected from all three study areas during the fall sampling season (September), when it was expected that the soil would be driest. A portable device (PI-SWERL^®^), was used to generate wind and simulate windblown dust emissions from test surfaces adjacent to where bulk soil sampling during previous visits provided positive samples for the fungus. The device has been described in detail elsewhere [67,68,69,70]. Briefly, the instrument consists of a cylindrical chamber that is open at the end (diameter = 30 cm) that is placed on the test soil surface. Wind shear is simulated by the rotation of an annular blade within the chamber and the resultant dust emitted from the test surface is vented through an exhaust port. For these tests, the PI-SWERL**^®^** was operated at a constant rotation rate of 5000 revolutions per minute, corresponding roughly to a surface wind speed of 15–20 m·s^−1^ (35–45 mph). Dust emitted from within the PI-SWERL^®^ was sampled through a PM10 (Particulate matter, 10 μm or smaller in aerodynamic diameter) size inlet that was connected to the exhaust port and subsequently onto a filter (Figure 5A). The collected dust on the filters was tested for *Coccidioides* DNA using the same methods used to detect the pathogen in soils. Material collected in the hopper of the size selective inlet (nominally only particles larger than 10 μm in aerodynamic diameter) was collected as well, and processed similarly, but separately.

In addition to sampling for the windblown component of dust with the PI-SWERL**^®^**, a vehicle-based mobile monitoring platform (TRAKER™, [67,71,72] was used to collect dust that is suspendable by travel on the unpaved roads (Figure 5B). Several samples of dust suspended in the wake of the TRAKER^TM^ test vehicle were collected for subsequent analysis during the fall sampling season. The intent here was to determine if exposure to vehicle suspended dust might be a pathway for exposure to arthroconidia not to try to pinpoint where *Coccidioides* DNA was detected in the greatest quantity. As with PI-SWERL**^®^** dust samples, the PM10 material collected on filters was analyzed separately from the material in the cyclone hopper.

In general, it was more challenging to collect sufficient sample material from sampling road dust with the TRAKER^TM^ than wind-suspendable dust from PI-SWERL**^®^**. This should not be taken as an indicator that wind erosion plumes are somehow more intense than road dust plumes. It is just that sample collection with the TRAKER^TM^ takes place in a part of the plume that has undergone more dilution than in the case of the PI-SWERL**^®^**. Road dust samples were collected over several miles of roadway travel, so that they do represent an average sample over spatial scales that are in some sense much more representative than can be collected with point measurements using the PI-SWERL**^®^**.

The risk of exposure to the pathogen might be linked to the type of environment that is disturbed due to human or natural influences. Therefore, all sampling sites were separated into two categories based on human influence: agricultural fields and orchards under management (category 1) and environments with lesser human influence, dormant agricultural fields or meadows, natural or semi-natural area with mainly Creosote and Salt Bush, grassland, or eroded land, and dirt roads (category 2). We then determined the risk of exposure to the pathogen based on the number of *Coccidioides* DNA-positive samples in each category (see Section 2.2).

### 2.3. Detection of Coccidioides spp.

DNA was extracted from all soil samples (two replicates) using the MoBio Powersoil DNA extraction kit (MoBio Laboratories, Carlsbad, CA, USA) following the protocol provided by the manufacturer. Prior to extraction, aliquots of soil were heated for 30 min at 70 °C and then incubated at 56 °C with Proteinase K (100 μg/mL) to enhance cell lysis from dormant microbes and thus, to increase the sensitivity of the culture-independent approach. Two different diagnostic primer pairs that were rather specific to *Coccidioides* were used to test for the presence of the pathogen’s DNA.

Briefly, to detect *Coccidioides* DNA, a nested PCR originally published by Baptista-Rosas et al. (2012) was used with modifications. This nested PCR included three individual PCR reactions to detect *Coccidioides*. In a first PCR, primer pair NSA3/NLC2 amplified a ~1000 bp 18S rDNA fragment of all fungi, followed by a second PCR using primer pair NSI1/NLB4 which is specific to Ascomycetes and some Basidiomycetes (~900 bp). An aliquot of a 1:25 dilution of the second PCR was then used in two diagnostic PCRs performed with primer pair ITS1Cf/ITS1Cr (~120 bp) [73] and with primer pair EC3/EC100 (~500 bp, [74]) which are both more specific to *Coccidioides* compared to the one originally suggested by Johnson et al. [10]. These diagnostic primer pairs target the intertranscribed region 1 and intertranscribed region 2 of the ribosomal gene, respectively. Two different diagnostic primer pairs were used to reduce potential PCR bias and other pitfalls of PCRs, such as inhibition or formation of PCR artefacts [75] to increase the potential of detecting DNA of the pathogen in soils that differ in organic matter content and other chemicals that might interfere with downstream PCR reactions if co-extracted. A positive control used for all PCRs was obtained from *C. posadasii* Δchs5 strain (NR4548, BEI Resources). Positive diagnostic PCR products were sequenced (Laragen Inc., Culver City, CA, USA) and compared to entries in the GenBank nucleotide database [76,77] to ensure that they indeed represent *Coccidioides* spp. It should be noted that detecting DNA of the pathogen in soil or dust samples confirms its presence but does not necessarily indicate that *Coccidioides* is active or infective.

### 2.4. Soil Analyses

Texture and ion chemistry were analyzed for soil samples using established techniques described below, such as grain size analysis, ion chromatography, pH and electrical conductivity analysis, at the laboratories at CSU Bakersfield and the Environmental Analysis Facility (EAF) at the DRI. Additionally, local site characteristics such as apparent disturbance due to human influence and vegetative cover were noted in the field. Disturbance and human influence was assessed visually; for example, if a site was converted to an agricultural field, orchard or meadow, invasive plant species were detected, and native plant species were absent or rarely documented, the site was ranked as disturbed. In contrast, sites that were dominated by native plant species were considered natural or semi-natural.

The percentage of sand, clay and silt in all soil samples (texture analysis) was determined at the Desert Research Institute (DRI) Soils Analysis Lab using a Malvern Mastersizer 2000 particle size analyzer. For pH measurements, we used a pH electrode (Oakton pH 510 series) and prepared a 1:4 ratio of soil:water to improve the fluidity of the slurry; suitable particularly for soils with high clay concentrations. Soil slurries were allowed to rest for four days to analyze total dissolved salts and electrical conductivity. Inorganic ion composition from a subset of samples selected from *Coccidioides* DNA-positive and -negative soils were obtained using the standard ion chromatography techniques employed by the (EAF lab using a Dionex ICS-3000 (Sunnyvale, CA, USA) ion chromatograph. Analytes included fluoride, chloride, bromide, nitrite, nitrate, sulfate, lithium, sodium, ammonium, potassium, magnesium, and calcium.

### 2.5. Statistical Analyses

Statistical analyses of environmental parameters, such as soil ions, and grain size, were performed using a permutation-based multivariate analysis of variance with Site and *Coccidioides* DNA presence/absence as independent variables in R [78]. A distance matrix using Bray–Curtis distances was built using the vegdist command, and this matrix was subsequently analyzed using adonis, both in the vegan package. Soil pH and electrical conductivity were analyzed independently via 2-factor ANOVA to investigate any significant interaction found.

We ensured that data used for these tests were normally distributed with the mean and median approximately equal, and with 68% of the data falling within one standard deviation. We investigated data variance using Levene’s test for equality of variance and identified data outliers (outside the range of the first and third quartile) using the program Excel and its data analysis tools prior to performing statistical data analyses.

In addition, we performed single and multiple regression analyses to investigate a connection between PM10 and precipitation on disease incidence using R [78]. Lastly, we compared environments that were severely impacted by human activities to environments with a lesser human impact regarding the presence or absence of the pathogen’s DNA by using the Chi-square test with Yates correction [79].

## 3. Results

### 3.1. Detection of Coccidioides in Soil and Dust Samples

Overall, 389 soil samples were investigated for the presence of pathogen DNA. *Coccidioides* DNA was detected in soil samples collected near all three military installations. Table 2, Table 3 and Table 4 indicate soil types, soil map unit numbers, coordinates and brief site descriptions, as well as results of diagnostic PCRs for each sampling location.

Figure 6 shows examples of PCR results obtained with both *Coccidioides*-specific primer pairs. Sequencing of PCR products revealed that DNA from both species *C. posadasii* and *C. immitis* was present in our sampling area. All PCR amplicons were 98–100% related to *C. immitis* (MK577425 (NAS Lemoore and Antelope Acres)) or *C. posadasii* (MK577421, MN520605 (MCAGCC Twentynine Palms)) entries in the GenBank nucleotide database. Table 5 shows a summary of all diagnostic PCR results for bulk soil samples. Overall, results obtained with two different diagnostic primer pairs agreed between ~67 and 85% depending on sampling area. PCRs with primer pair ITSC1f/r which amplifies a short fragment of ~120 bp seemed to be superior in sensitivity (8.6% more positive PCR amplicons confirmed via sequencing), compared to PCRs with primer pair EC3/EC100 which amplifies a ~500 bp amplicon. Primer pair ECf3/EC100r however, occasionally amplified DNA from unrelated soil fungi, most often *Cladosporium* spp. which comprised approximately 4% of the amplicons that were sequenced. Even though the ITSC1f/r primer pair seemed to be superior to the EC3f/EC100r PCR regarding sensitivity and specificity, a longer amplicon can be used for phylogenetic analyses of *Coccidioides* spp. [74]. Results from replication of PCRs from two different DNA extracts per soil sample agreed well (97.2% agreement).

Using the PI-SWERL**^®^**, we collected two wind-suspendable dust samples west of Edwards AFB, four from around NAS Lemoore, and seven from near Twentynine Palms MCAGCC. Using the TRAKER^TM^, we collected one sample of resuspended road dust from around Antelope Acres, two samples from NAS Lemoore, and two samples from around Twentynine Palms. We detected DNA of the pathogen in one sample from Lemoore (near site J, PI-SWERL**^®^**), a site that was dominated by Iodine Bush (*Allenrolfea occidentalis*), a plant that indicates a highly saline environment. We also detected *C. immitis* in dust collected near site 5 at Antelope Acres (PI-SWERL**^®^**), a highly eroded soil with notable presence of common rabbit brush (*Ericameria nauseosa*) and occasional Salt Bush (*Atriplex* spp.). *Coccidioides posadasii* was detected in five dust samples collected near Twentynine Palms (sites 5, 8–10, PI-SWERL**^®^** and TRAKER^TM^), at sites dominated by Creosote (*Larrea tridentata*) and occasional Salt Bush (*Atriplex* spp.) (Table 6).

In addition, several soil core samples were collected from all three study areas in May/June and September/October. Not all sites could be sampled down to 30 cm depth because of underlying rocks. Table 7 shows results of the diagnostic PCRs for all soil core DNA extracts. *Coccidioides* DNA was predominantly detected near the soil surface. Soil core analyses revealed that DNA of the pathogen was present in soil layers down to 30 cm depth but was detected more frequently in the upper 10 cm of the soil (A and B horizons). Ten representative PCR amplicons obtained with primer pair EC3/EC100 were submitted to the GenBank database (Accession # MT436381–MT436390).

### 3.2. Environmental Analyses

The analyses of soil environmental parameters revealed quite distinct soils at each of our three main sampling locations. The soil particle data were analyzed via a permutation-based multivariate analysis of variance with site and *Coccidioides* DNA presence/absence as independent variables in R [78] with type III sums of squares A distance matrix using Bray–Curtis distances was built with the vegdist command using the gravel, sand, silt, clay and CaCo_3_ data, and this matrix was subsequently analyzed using adonis, both in the vegan package. This analysis showed no interaction between site and the presence of the pathogen (F_1,31_ = 2.1, *p* > 0.1) and no effect of *Coccidioides* presence (F_1,31_ = 0.42, *p* > 0.6) based on the detection of its DNA, but a significant effect of site (F_2,31_ = 21, *p* < 0.005). A post-hoc pairwise comparison performed between sites with the pairwise.adonis package with Bonferroni correction showed that Twentynine Palms and Antelope Acres did not differ (F_1,28_ = 2.2, *p* > 0.3), while the Lemoore site differed from both Twentynine Palms (F_1,23_ = 21, *p* < 0.01) and Antelope Acres (F_1,17_ = 17, *p* < 0.005) in the distribution of particle types. Figure 7 shows that the major driver of the observed differences in particle size are the reduced sand and increased silt concentrations in Lemoore soils compared to samples from the other two sites.

The remaining soil characteristics, pH and electrical conductivity were treated together because of the suspicion that they may be correlated, and so were analyzed using a multivariate permutation analysis parallel to the soil ion and particle size analyses. Electrical conductivity was found to have much larger variation in the data from Lemoore than from the other sites; so, the log of electrical conductivity was used for all analyses. This analysis resulted in a significant interaction between site and *Coccidioides* DNA presence (F_1,26_ = 9.3, *p* < 0.005), and significant effects of both *Coccidioides* DNA presence (F_1,26_ = 31, *p* < 0.001), and site (F_2,26_ = 7.9, *p* < 0.01). pH and electrical conductivity were analyzed independently via 2-factor ANOVA to investigate the significant interaction found.

Soil pH values appeared higher in samples collected around NAS Lemoore and Twentynine Palms with an average in both cases of pH 7.81 compared to Antelope Acres west of Edwards AFB with an average of pH 7.23. The pH of soils in which DNA of the pathogen was detected ranged between pH 7.1 and 8.1 (Figure 8A). For pH, the statistical results paralleled those found earlier: no interaction between site and pathogen DNA presence (F_2,26_ = 0.90, *p* > 0.4), no effect of pathogen DNA presence (F_1,26_ = 0.005, *p* > 0.9), and a significant effect of site (F_2,26_ = 7.0, *p* < 0.005). A Tukey HSD considering only site showed that Lemoore soils had a higher pH than both Antelope Acres (difference = 1.04, *p* < 0.005) and Twentynine Palms (difference = 0.71, *p* < 0.05), but that Antelope Acres and Twentynine Palms did not differ significantly from one another (difference = 0.33, *p* > 0.2).

Electrical conductivity (Figure 8B), however, showed a significant interaction between site and presence of *Coccidioides* DNA (F_2,26_ = 11, *p* < 0.001). In this initial analysis, both presence of *Coccidioides* DNA (F_1,26_ = 39, *p* < 0.001) and site (F_2,26_ = 6.8, *p* < 0.005) showed significant results. Figure 8B details the interaction, showing that Lemoore soils had a higher electrical conductivity than Twentynine Palms, which is consistent with the patterns found before. The significant effect of the presence of *Coccidioides* DNA, however, appears to be an artifact of the interaction. Although there were no differences within any single site, the sites without *Coccidioides* DNA in both Twentynine Palms and Lemoore soils appeared to be slightly lower in conductivity than *Coccidioides* DNA-positive sites, while this tendency is reversed for soils from Antelope Acres. The small sample sizes and correspondingly large error bars likely caused this rather discrepant result.

Ion chromatography was performed for a subset of *Coccidioides* DNA-positive and -negative samples from all main sampling areas to determine differences in soil ion content. Results revealed distinct fingerprints of the soil ion content for the different regions investigated. The ion chromatography data were analyzed via a permutation-based multivariate analysis of variance with site and *Coccidioides* DNA presence/absence) as independent variables in R [78] with type III sums of squares. All values were transformed as ln(value + 1) before analysis to deal with large variation in magnitudes of values, including some zeroes. A distance matrix using Bray–Curtis distances was built using the vegdist command, and this matrix was subsequently analyzed using adonis, both in the vegan package. This analysis showed no interaction between site and pathogen DNA presence (F_1,35_ = 0.47, *p* > 0.7) and no effect of *Coccidioides* DNA presence (F_1,35_ = 1.3, *p* > 0.2; Figure 9), but a significant effect of site (F_2,35_ = 29, *p* < 0.001). A post-hoc pairwise comparison was performed on just the site differences using the pairwise.adonis package with Bonferroni correction. This shows that all sites differed from one another in the distribution of ions in the soil: Twentynine Palms versus Antelope Acres (F_1,32_ = 3.2, *p* < 0.05); Twentynine Palms versus Lemoore (F_1,24_ = 49, *p* < 0.005); Antelope Acres versus Lemoore (F_1,20_ = 64, *p* < 0.005). Figure 9B shows that Lemoore is clearly richer in all ions than either of the other two sites, while the differences between Antelope Acres and Twentynine Palms are generally minor in comparison. Soil samples near NAS Lemoore, which is situated in an agricultural area, where only remnants of natural vegetation remain, contained high levels of ions such as sulfate (65 mg/g averaged) and sodium (25 mg/g averaged) likely due to soil supplementation with fertilizers compared to soils from the Mojave Desert (0.18 and 0.12 mg/g, averaged respectively. Furthermore, elevated levels of calcium and chloride (6 and 5 mg/g, averaged respectively) were detected in soils from the Lemoore area. Soils collected near Twentynine Palms, a non-agricultural environment, exhibited much lower levels of the aforementioned ions. Results from soils collected near Antelope Acres were similar in ion composition to those from the Twentynine Palms area but showed a slightly higher content of nitrate and significantly less soluble Calcium. The major ions detected in these desert soils were soluble potassium and calcium, likely originating from granite and basalt which are major soil parent material in the Mojave Desert. Potassium ions were also present in soils from Lemoore and Antelope Acres in similar concentrations. However, Ca content was significantly higher in soils collected from the Lemoore area. Bromide and soluble Lithium were generally absent from soils collected near Twentynine Palms but were present in low amounts in soils from areas influenced by agriculture. Significant differences in soil ion content were observed between the three main sampling locations, (Figure 9A).

Overall, *Coccidioides* DNA was most often detected in natural or semi-natural desert environments that were not heavily influenced by humans. Natural areas showed at least some California-native vegetation, including Salt Bush, Creosote and native annuals. In contrast, human-influenced environments (agricultural areas and orchards under management) were generally negative for the pathogen’s DNA, as well as grassland dominated by non-native annuals. This was investigated by pooling sites into the above mentioned two categories. Relatively undisturbed sites had a higher prevalence of *Coccidioides* DNA than highly disturbed sites (Chi-square test with Yates correction = 23.2, df = 1, *p* = 1.4 × 10^−6^). (Table 8 and Figure 10). Based on our findings, we propose that *Coccidioides* DNA-positive sites, if not disturbed might not pose a high risk of pathogen exposure and infection (Figure 11).

Several hot spots of the pathogen that were detected in this study, based on the presence of its DNA, should be highlighted. One site that was revealed as a potential growth site of the pathogen in the Lemoore area was located near the intersection of Jackson Road and Highway 41 with daily traffic connecting Lemoore and Stratford (Kings County). The sandy loams in this area (landform: alluvial fans and rims on basin floors) are highly-saline with elevated pH, have an elevated electrical conductivity and total dissolved salts, and are named alkali-sinks. This location showed substantial disturbance by off-road vehicles, a potential dust-hazard in the dry season that increases the risk of being exposed to arthroconidia of the pathogen for humans and animals in this area. Satellite imagery (e.g., google earth and landsat) reveals that the general area southeast and east of NAS Lemoore is composed of salty sandy loams (Figure 12). Most of this land has been converted to agricultural fields in recent decades. California-native Iodine Bush (*Allenrolfea occidentalis*) was the dominant plant species present. This plant could be an indicator species of a suitable habitat for *Coccidioides* besides Salt Bush in this area. However, based on results of this study, this location in general has a lower risk of pathogen exposure for military personnel and the public compared to the area near EAFB and MCAGCC in the Mojave Desert. Highly eroded areas near Antelope Acres were identified as *C. immitis* habitat (Figure 13) and Creosote dominated land west and south of MCAGCC Twentynine Palms was identified as a highly endemic area of *C. posadasii* (Figure 14). The general distribution of plant species that could serve as indicator species for an environment suitable for *Coccidioides* growth throughout California can be seen on an interactive map provided by Calflora.org, https://www.calflora.org/cgi-bin/species_query.cgi?where-calrecnum=171.

A linear regression analysis was performed in order to estimate the relationship between precipitation and PM10 concentration and incidence of coccidioidomycosis between 2002 and 2018, as obtained from the Centers for Disease Control and Prevention (CDC). The slope for the linear trend between PM10 concentrations and coccidioidomycosis incidence was 0.4985 (95% CI, F_1,14_ = 1.37, *p*-value > 0.25). The slope for the linear trend between precipitation and incidence was 0.0748 (95% CI, F_1,15_ = 0.02, *p*-value > 0.89). The *p*-value improved to 0.47 when a one-year delayed effect of precipitation on disease incidence was considered, but results were still not significantly different. 

A multiple regression, combining the effects of PM10 and precipitation was also performed, and was found not significant, as well (95% CI, F_3,12_ = 0.73, *p*-value > 0.5). Considering a delayed effect of precipitation and PM10 on coccidioidomycosis incidence by one year was still not significant in a multiple regression analysis (95% CI, F_3,11_ = 2.609, *p*-value > 0.1). PM10 data were obtained from the California Air Resource Board (https://www.arb.ca.gov/adam) and precipitation data were obtained from the National Oceanic and Atmospheric Administration (NOAA) National Centers for Environmental Information (https://www.ncdc.noaa.gov/cag/divisional/rankings/0407/pcp/200201).

## 4. Discussion

Selection of the study areas was driven by the objectives of the project, which had a focus on impacts on military personnel. However, areas around military facilities also have two advantages compared to choosing random areas around population centers from a map. First, the Department of Defense keeps detailed statistics by year and location for disease incidence and morbidity by cause. Second, because military facilities are well delineated and military life centers around working and often living in the immediate vicinity of such facilities, disease incidence could be tied to exposure in a specific geographic region with greater confidence. All in all, these two factors provide greater confidence that disease incidence reported at a specific military facility was likely a result of exposure within or in the immediate vicinity of the facility. It is noteworthy that incidence among prison inmates in some parts of California [80] also provides a direct link between specific location of exposure and location of disease that is more difficult to establish in more regional outbreaks. Areas around prisons with reported cases of Valley fever would make excellent future sites for examination of the link between activity and exposure.

Even though coccidioidomycosis is prevalent at all three of our study areas, the landscape is quite different. However, in all locations fine-grained saline-alkaline soils (quaternary alluvium, sandy loams and clay loams) are common. Agricultural management in the San Joaquin Valley has changed the landscape significantly over the last 100 years. Originally desert shrubs such as those found in the Mojave Desert were also common in some areas of the Central Valley, especially in the southwest [81]. Samples were collected from publicly accessible locations near these facilities along transects that cover different soil types as indicated by the United States Department of Agriculture (USDA) websoilsurvey (WSS) database. By randomly sampling along informal transects, we were able to obtain samples that differed regarding degree of human influence, soil physical and chemical parameters, and vegetation cover.

The data we obtained in this study did not allow us to link individual environmental parameters to the presence or absence of the pathogen in the soil. However, it became evident that natural or semi-natural environments that support small mammal communities and other wildlife and that are characterized by no or only a moderate degree of human influence, were often confirmed as *Coccidioides* habitats. In contrast, agricultural soils and orchards under management rarely tested positive for the pathogen. The presumed risk of exposure to the pathogen did vary among study areas, with the area near MCAGCC Twentynine Palms showing the greatest number and percentage of *C. posadasii* DNA-positive sites compared to the environment around NAS Lemoore which is known for industrial agriculture and where *C. immitis* DNA was detected in remnants of natural areas.

The results of the present study have refined our conceptual understanding of the exposure pathway to *Coccidioides.* It appears that the growth sites of the pathogen are mostly restricted to comparatively undisturbed areas where vegetation such as Creosote and Salt Bush are still in place. This is true for the sampling sites around NAS Lemoore, Edwards AFB, and MCAGCC Twentynine Palms. Perhaps this is because more natural, less disturbed areas are preferred by wildlife, that in turn provides a source of rodent keratin that can be accessed by the fungus [13,82] in soils that are otherwise low in organic matter content. Landscapes that are disturbed by being reworked to the point of becoming an agricultural field are apparently rendered inhospitable for *Coccidioides*, possibly due to lack of the right combination of organic material in the soil (e.g., rodent keratin), the application of chemicals that inhibit fungal growth, the physical disruption caused by frequent soil disturbance such as tilling, the establishment of microbes that act as antagonists to the pathogen, or some combination of these factors [83,84,85,86,87,88].

We hypothesized that if soil that is hospitable for *Coccidioides* growth remains undisturbed, the pathogen might only rarely or in small numbers become airborne under the influence of wind alone. This is supported by the fact that PI-SWERL^®^ testing on sites that were known to be positive for the fungus (based on bulk samples collected from the first few cm of the soil surface) resulted in suspended dust samples that were largely negative except from locations that were along the boundary of a vegetated area and a non-vegetated area where the potential for wind erosion is much higher. Possibly, the same parameters that make a location hospitable for the growth of *Coccidioides* (native shrubs and annuals, rodents, BSCs) are also parameters that on balance reduce the wind erodibility of the surface. If true, this might suggest that simply standing in a landscape where *Coccidioides* is growing over a large areal extent may not be much of an exposure hazard. Of course, there is certainly some exposure hazard because presumably the fungus relies on the aeolian transport of arthroconidia to propagate itself, but surprisingly, this might be a relatively minor pathway for exposure. The analyses of soil core samples also revealed that the pathogen is established in deeper soil layers (at least 5–25 cm), based on the detection of its DNA, and does not exclusively reside on the soil surface where it is more likely to encounter high temperatures, desiccation and UV radiation. It would be very useful to learn with greater precision the distribution of the pathogen within the first few centimeters of the soil surface also in response to environmental changes over the seasons.

The soil sample analyses indicated that elevated pH (7.1–8.1), electrical conductivity, total dissolved salts, as well as silt and clay content can be helpful in describing a suitable *Coccidioides* habitat, but none of these appeared to be first order parameters in controlling the occurrence of the fungus. Our findings indicate that pH, electrical conductivity and total salts alone can vary over a relatively large range and cannot be used as specific indicators of suitable habitats that allow *Coccidioides* species to grow. However, there is indication that an increase in fine particles (silt and clay) elevated CaCO_3_ and soluble Ca and K, as well as reduced levels of sulfate and sodium in soil could be used as a marker for higher probability of *Coccidioides* habitats. The presence of sparse growth of halotolerant xerophytes such as *Atriplex* spp., *Allenrolfea occidentalis*, and *Larrea tridentata* can indicate these soil conditions.

At NAS Lemoore and Antelope Acres near EAFB, the two road dust samples (TRAKER^TM^) were negative for *Coccidioides* DNA. At Twentynine Palms, the road dust sample that was collected from roads in the area surrounding sites 29P7-10 was positive for *C. posadasii* DNA with both diagnostic primer pairs. It is difficult to place this limited information into any kind of quantitative assessment of exposure risk. This is especially true because a positive DNA test does not provide any information about any risk of infection, because it is not known if the pathogen’s arthroconidia are viable in a human host. At the simplest level, these findings suggest that traveling on unpaved roads in endemic areas is a potential pathway for exposure for vehicle passengers and those that are downstream of the road dust plume. In the specific case of military personnel, given the extensive exposure to road dust that troops are subjected to during training activities (especially when in convoys) [89], this may be a significant pathway for infection [37].

Given that evidence of the fungus was found to depths of 25 cm and the likelihood that viable arthroconidia are more prone to be found at some depth where UV radiation and temperature are more favorable for microbial growth, we hypothesize that exposure likely happens in the immediate vicinity of land disturbance activity at the time of the activity and probably for some time after the disturbance. This can include activities such as trenching, grading, traveling on the surface with tracked vehicles, or any activity that significantly abrades or punctures the soil crust to the point that inhalable arthroconidia are physically lifted out of and around growth sites into the air in the vicinity of humans. This is likely the type of exposure that one would experience during varied activities such as construction (e.g., solar plants) [90], driving tent stakes into the ground, and walking behind a tracked vehicle as it travels on a relatively undisturbed surface for the first time [91]. What is not at all clear is how long site disturbance causes an increased risk of inhalation and how far the increased risk propagates.

Our hypothesized prevalent exposure pathway does not preclude the possibility that exposure also happens over a more diffuse area during widespread wind erosion events (such as during frontal passage) or during travel on unpaved roads in *Coccidioides*-endemic areas. Indeed, our measurements showed positive samples for *Coccidioides* DNA both from wind suspended dust near growth sites and from vehicle suspended dust while driving through and in the vicinity of those sites. However, given that the fungus has apparently widespread presence at the three study areas and that wind events are quite frequent in the southwestern U.S., one would expect that the occurrence of illness would be much less clustered in space (in contrast to outbreaks in prisons, among construction workers, within military personnel that were deployed together) and much more clustered in time specifically following discrete wind events. It seems instead more plausible that when a growth site is disturbed and active growth habitats at some depth are brought to the surface, this is what creates a true surface “hot spot” area, where large amounts of viable, potentially infectious material are liberated and are either lofted locally by earth-moving activities and/or transported some distance by wind.

The larger vegetation in the Mojave Desert is generally scattered due to the limited amount of water available. Drought tolerant plants such as Creosote (*Larrea tridentata*) and Salt Bush (*Atriplex* spp.) are common in endemic areas of the pathogen. In the area between plants, Biological Soil Crusts (BSC) can be encountered that are not only important for the nutritional status of desert soils, but BSCs also enhance soil quality by aggregating soil particles, thereby reducing wind and water erosion [92,93]. However, extensive disturbance poses a risk of covering or destroying BSCs, preventing photosynthesis, and ultimately causing their degeneration and death. BSCs need many years to develop and once destroyed can be difficult to rehabilitate, leaving bare soil behind, vulnerable for further erosion [94,95]. Remote sensing of Biological Soil Crusts in the Mojave Desert revealed a substantial loss of cyanobacterial crusts and lichens [96]. It is possible that degradation of BSC in areas where the fungus is found can enhance the possibility of exposure due to high wind events.

There are several limitations of our study that should be addressed. With regard to suspended road dust, the limited number of samples collected does not allow for in-depth understanding of the relationship between road dust and exposure to *Coccidioides*. However, road dust from the Twentynine Palms area did test positive for the fungus, confirming that simply traveling on unpaved roads in endemic areas is a potential pathway for exposure, possibly a very important one for individuals and groups that may have occupational-level exposure to road dust (e.g., troops during training activities).

One parameter that was not investigated in the present study but that has promise as a determining factor in whether a growth site is suitable for the pathogen is the type of organic matter that is present in the soil. As a species that is less able to biodegrade plant derived organic matter, and preferentially uses animal derived matter such as keratin, high rodent abundance that was observed at sampling sites in the Mojave Desert could be included as an indicative parameter [13,97,98].

The conceptual model for exposure that we have outlined, if shown to be true, can result in tools that are immediately useful for minimizing exposure to the fungus. Landsat satellite imagery can be used to identify large highly degraded locations or ongoing and planned construction within the setting of endemic areas of the pathogen that together with wind data indicate high risk areas for *Coccidioides* exposure. Maps can be retrieved for past years as well, which allows the documentation of changes in risk of exposure to the pathogen due to changes in environmental conditions including human activities. Together with more information about the lifecycle of the fungus, these tools can be used for guided management decisions about certain activities and provide actionable prophylactic guidelines. For example, it is possible that exposure to the fungus from travel on unpaved roads could be greatly reduced by adding a buffer between the travel portion of the road and the adjacent vegetation. Some basic guidelines about the level of activity to be performed and the use of Personal Protective Equipment (PPE) could be provided for potential occupational exposure reduction (construction, grading, etc.). Immunosensitive persons could be notified if excavating activities are to occur nearby and provided guidance on how long they might pose a risk.

These ideas suggest a path forward for future applied research on this topic. The following components are among those that ought to be included in future work:Verify that the maximum exposure hazard is at the nexus of the somewhat ubiquitous (but highly dispersed) growth sites and activities that result in physical resuspension (e.g., earth-moving activities) or Aeolian resuspension. An accurate depth profile of *Coccidioides* is likely to be very informative for this purpose.Use large spatial datasets to define elevated conditions for exposure with the focus on the development of a *Coccidioides* early warning system, which, if implemented, will have a direct impact on public health.Investigate how climate-related events like drought and wildfires which are increasing in *Coccidioides*-endemic areas, are linked to fugitive dust development and coccidioidomycosis outbreaks in endemic and non-endemic areas of the pathogen.Characterize geographic risks, particularly in the context of environmental change, identifying further risk reduction strategies for high-risk groups.

Prevention or reducing the incidence of Valley fever can only be accomplished when the ecology of *Coccidioides* is known and when the complex interplay of environmental parameters that support or inhibit the growth of the pathogen now and in the future is better understood, followed by education of those who are actively contributing to soil disturbance and fugitive dust emissions, including the military and land developers. Our work contributes to a better understanding of *Coccidioides* exposure risks in different areas of the endemic region of this fungal pathogen by proposing a conceptual model and by proposing foci of future work.

## 5. Conclusions

This study showed that military personnel stationed at bases in the endemic area of *Coccidioides* are at risk of pathogen exposure from dust originating from disturbed soils in the area. However, the results of the present study have refined our conceptual understanding of *Coccidioides* exposure. Our work showed that sites where *Coccidioides* DNA was detected are largely restricted to areas where California-native vegetation such as Creosote, Salt Bush or BSCs are still in place. The same parameters that make a location hospitable for the growth of *Coccidioides*, such as intact native vegetation cover are also parameters that reduce wind erodibility of the surface. If true, this might suggest that simply walking on a windy day through a landscape where *Coccidioides* is growing over a large areal extent may not be itself much of an exposure hazard—in contrast to freshly disturbed natural areas. Preventing a further increase in coccidioidomycosis incidence in California requires improved dust mitigation management when soil is disturbed during construction in pristine desert lands, as well as expert environmental consulting on land management in general, which includes soil testing for the pathogen prior to soil disturbance in *Coccidioides*-endemic areas. We also recommend a minimization of land development in highly endemic areas of the pathogen that are relatively pristine to avoid a further increase in coccidioidomycosis incidence in the future.

## Figures and Tables

**Figure 1 ijerph-17-05285-f001:**
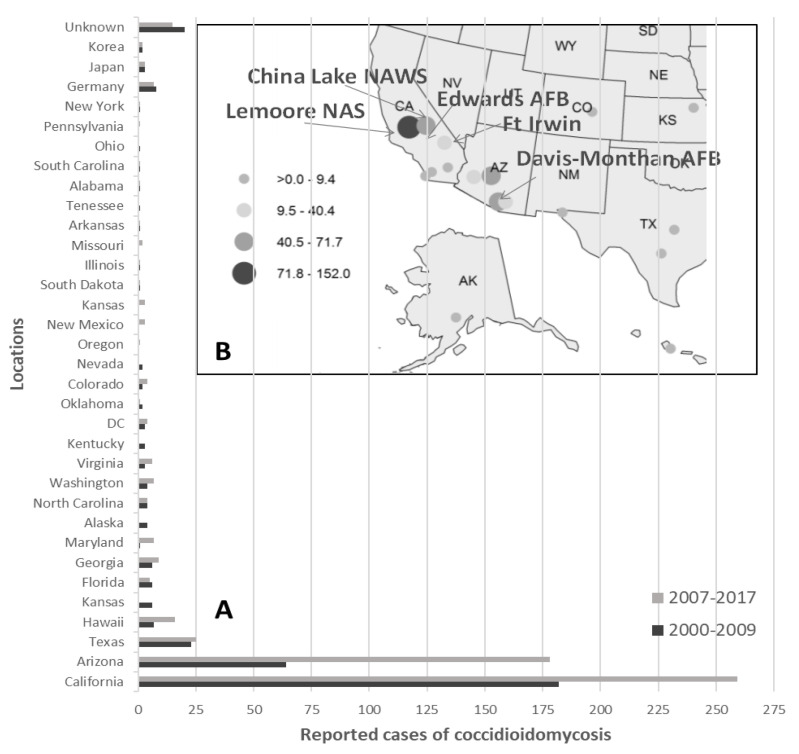
Number of diagnosed cases of coccidioidomycosis between 2000 and 2017 among active military. Data were compiled from two studies: Medical Surveillance Monthly Report (MSMR) 2010 Vol. 17(12):13 [25], and MSMR 2018 Vol. 25(4):2–5 [29] (**A**). Incidence of coccidioidomycosis within the active component, U.S. Armed Forces, 2000–2013 per 100,000 persons/year in the southwestern U.S. (adopted from MSMR 2014, Vol. 21(16):12–14 [26]) (**B**).

**Figure 2 ijerph-17-05285-f002:**
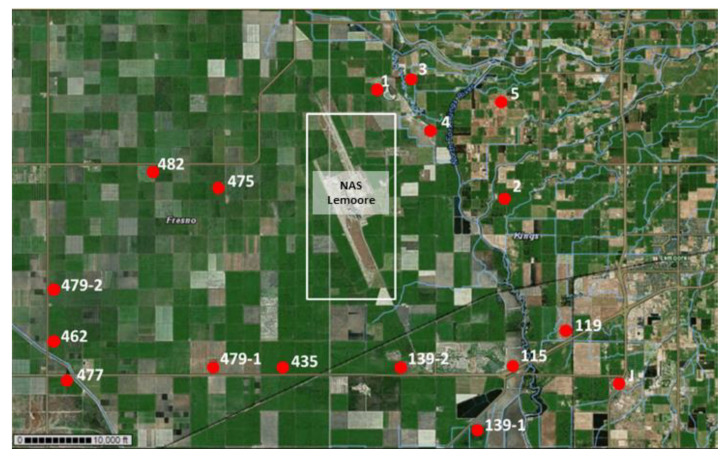
Sampling locations (indicated as red dots) around NAS Lemoore (in the center indicated within white rectangle) in Kings County and Fresno County, CA. Several subsamples were collected at each indicated sampling site (site 151 is off the map, south of site 139-1).

**Figure 3 ijerph-17-05285-f003:**
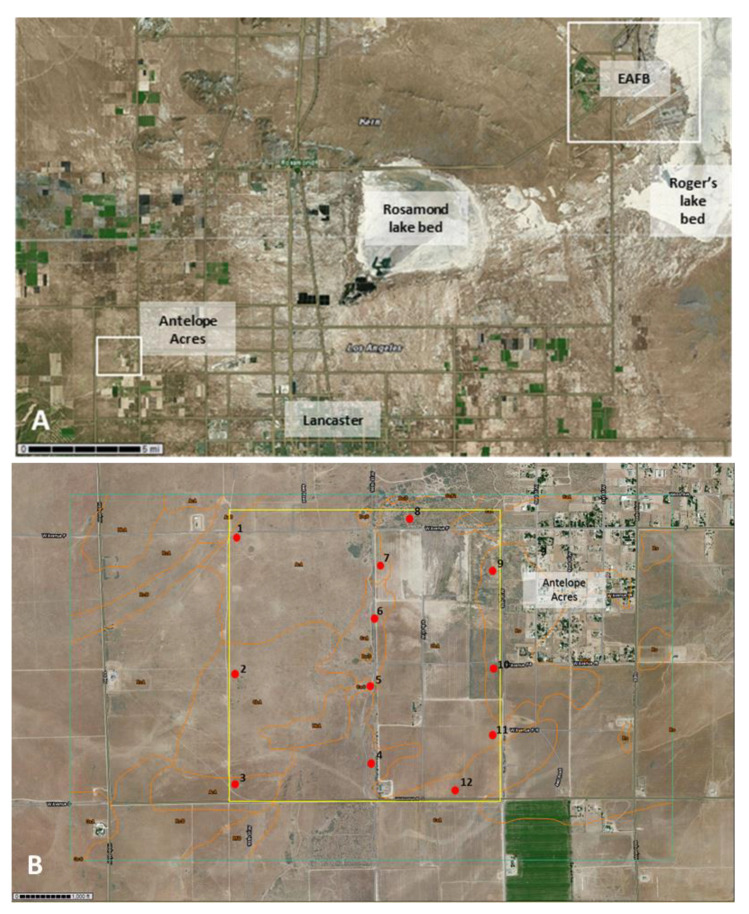
Sampling area in the western Mojave Desert where EAFB is located (upper right corner within the white square). Rosamond dry lakebed (center) and Roger’s dry lakebed (upper right) can be seen. The sampling location is indicated with a smaller white square southwest of Rosamond dry lakebed (**A**). The sampling area west of Antelope Acres with individual sampling locations are indicated as red dots (**B**). Several subsamples were collected at each indicated sampling site. Orange lines indicate borders of different soil types (United States Department of Agriculture (USDA) websoilsurvey (WSS) database).

**Figure 4 ijerph-17-05285-f004:**
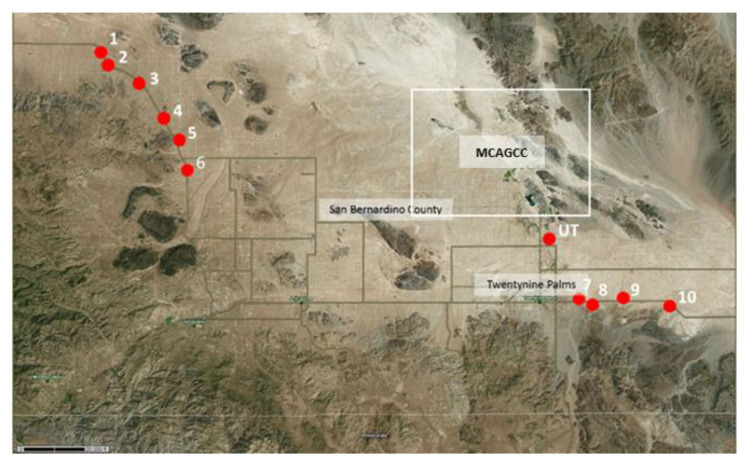
Sampling location (indicated as red dots) near Twentynine Palms MCAGCC (within white square). Several subsamples were collected at each indicated sampling site (site UT = Utah Trail).

**Figure 5 ijerph-17-05285-f005:**
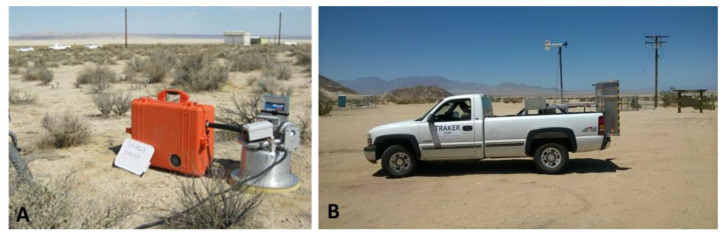
PI-SWERL**^®^** includes a cylindrical chamber that allows for recreating surface wind shear and measuring the resultant dust emissions. The orange case contains filter sampling apparatus with PM10 size selective cyclone (**A**). TRAKER^TM^ was developed as a tool to measure emissions of particulate matter as a result of vehicular travel on unpaved roads (**B**).

**Figure 6 ijerph-17-05285-f006:**
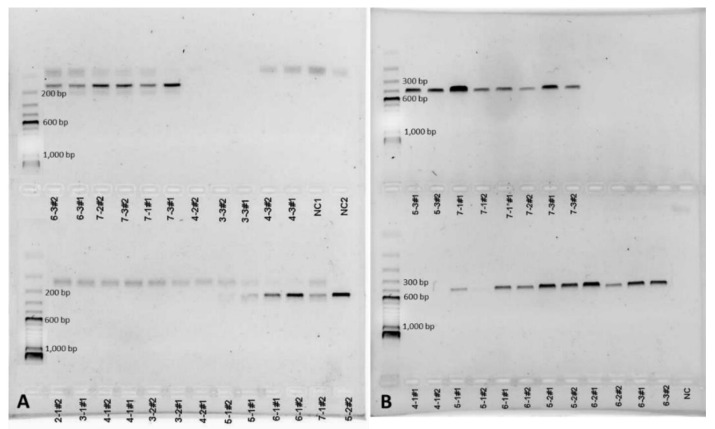
Examples of PCR results obtained with both diagnostic primer pairs using soil DNA extracts (5–7 cm depth) from Antelope Acres west of EAFB. Amplicons obtained with primer pair ITSC1Af/r (~120 bp) (**A**). Amplicons obtained with primer pair EC3/EC100 (~500 bp). A 100 bp DNA ladder can be seen in the 2% agarose gel on the left (**B**) (PC = positive control, NC = negative control).

**Figure 7 ijerph-17-05285-f007:**
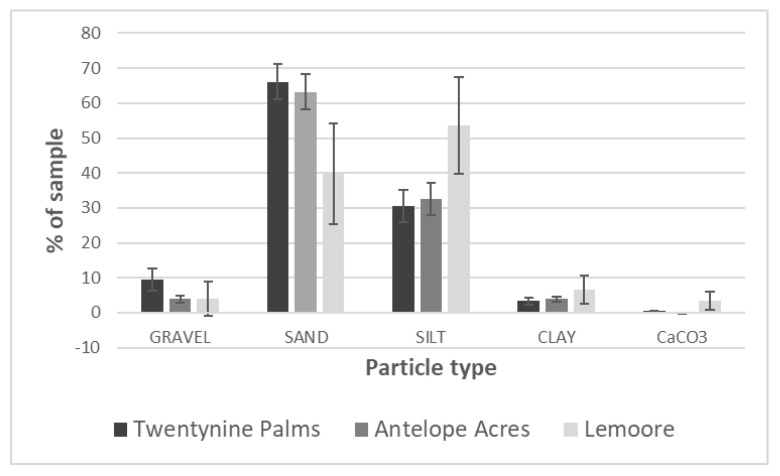
Comparison of grain size analyses of representative soil samples from all three main sampling locations. The graph shows the percentage of each soil particle type in soils from the three locations investigated (*n* = 18 for Twentynine Palms, *n* = 12 for Antelope Acres, and *n* = 7 for Lemoore sites. Error bars represent 95% Confidence Intervals).

**Figure 8 ijerph-17-05285-f008:**
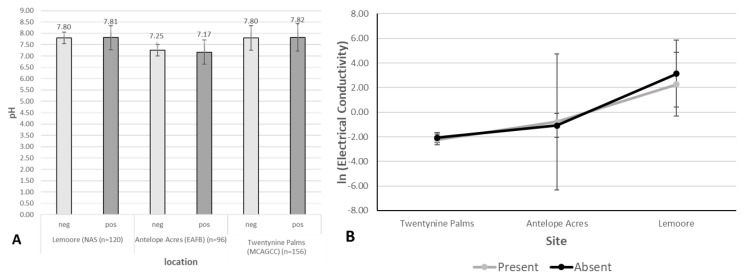
(**A**) Soil pH averages for *Coccidioides* DNA-positive samples (dark grey) and *Coccidioides* DNA-negative samples (light grey) from all the three locations (averages, error bars are based on 95% confidence). (**B**) Ln (Electrical Conductivity) by site and presence of *Coccidioides* DNA. Twentynine Palms: *Coccidioides* DNA present *n* = 10, *Coccidioides* DNA absent *n* = 8. Antelope Acres: *Coccidioides* DNA present *n* = 3, *Coccidioides* DNA absent *n* = 6. Lemoore: *Coccidioides* DNA present *n* = 3, *Coccidioides* DNA absent *n* = 3.

**Figure 9 ijerph-17-05285-f009:**
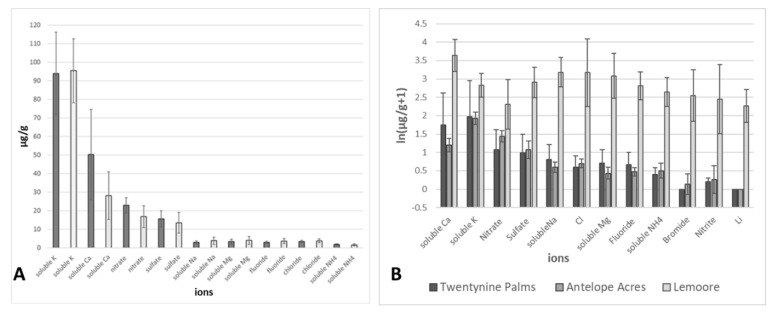
(**A**) Soil ion composition was not statistically different in soils where DNA of *Coccidioides* was detected (dark grey bars) and where DNA of the pathogen was not detected (light grey bars). (**B**) Results of soil ion analysis for selected *Coccidioides* DNA-positive and -negative samples form Twentynine Palms, Antelope Acres, and Lemoore (data: all sites combined: NAS Lemoore, Antelope Acres (west of EAFB), and MCAGCC Twentynine Palms, *n =* 41).

**Figure 10 ijerph-17-05285-f010:**
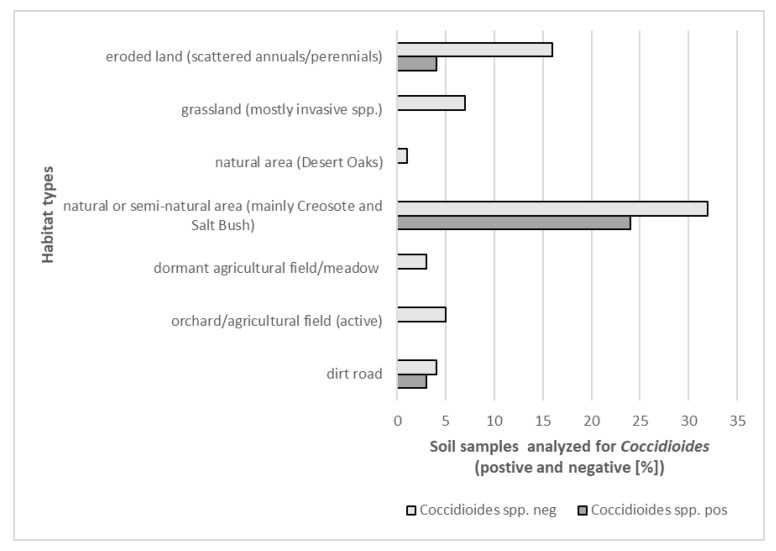
Detection of *Coccidioides* DNA in habitats that differ in the degree of human influence. Statistically significant differences were determined between different habitat types and the presence or absence of the pathogen DNA (*p*-values < 0.001, linear regression; see Methods and text).

**Figure 11 ijerph-17-05285-f011:**
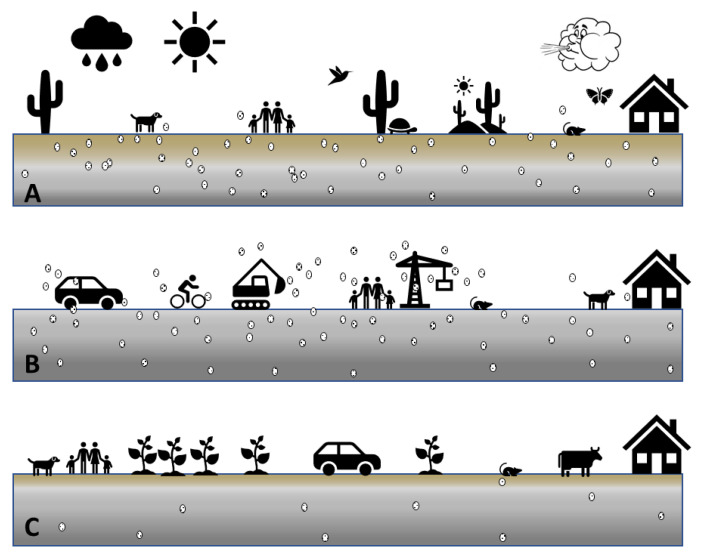
Conceptual model of environmental exposure pathways for Valley fever. (**A**) Undisturbed natural landscape with intact soil and diverse vegetation, including native species. *Coccidioides* growth supported, but few arthroconidia airborne (indicated as white, small, oval-shaped objects with black dots). (**B**) Freshly disturbed natural landscape with reduced or no vegetation. *Coccidioides* growth supported with increased airborne arthroconidia. Vehicle traffic, off-road biking, construction, as well as oil drilling are factors that highly erode soils. (**C**) Aged disturbed landscape with reduced or altered vegetation, such as agricultural fields under management, orchards with suppressed vegetation between trees, or dairy farms with high manure contamination of the soil. *Coccidioides* growth is inhibited and few arthroconidia become airborne.

**Figure 12 ijerph-17-05285-f012:**
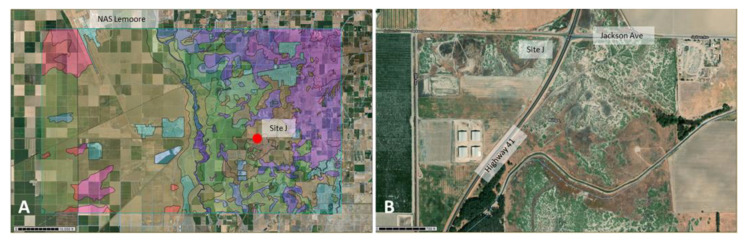
Location of *C. immitis* DNA-positive site J southeast of NAS Lemoore (red dot). Different soil types are indicated in colors with Boggs and Lemoore sandy loam in brown. These types of soils comprise ~6.5% of the area of interest (AOI) in this picture (**A**). Magnified satellite view of site J showing the extent of soil disturbance and its proximity to Highway 41 and Jackson Avenue, two highly frequented roads south of Lemoore (**B**).

**Figure 13 ijerph-17-05285-f013:**
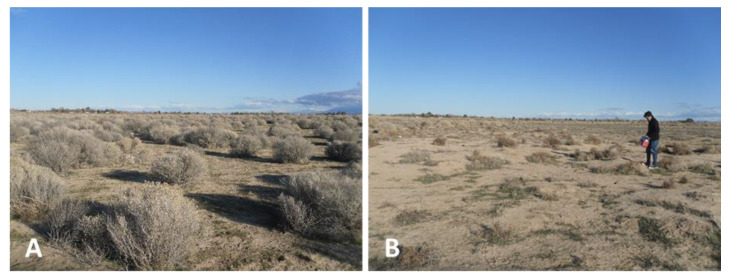
Sites 5 (**A**) and site 6 (**B**) west of Antelope Acres and EAFB where *C. immitis* DNA was detected, with scattered common rabbit brush and Salt Bush. Site 6 showed increased disturbance and erosion compared to site 5.

**Figure 14 ijerph-17-05285-f014:**
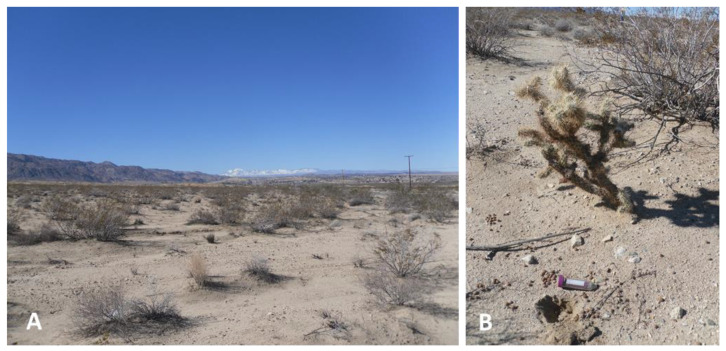
Overview of site 8 near Twentynine Palms showing scattered vegetation of predominantly Creosote (*Larrea tridentata*) (**A**). *C. posadasii* DNA-positive sampling site 9-3 near a Pencil Cholla (*Cylindropuntia ramosissima*) and Creosote in the background (**B**). Note the abundant rodent pellets nearby.

**Table 1 ijerph-17-05285-t001:** Overview of numbers of bulk soil (5–7 cm), soil core (0–30 cm at intervals), and dust samples collected around three U.S. military bases at different times of the year.

Soil Sampling					Dust Sampling	
Study Area	Bulk Soil Sampling			Soil Core Sampling	PI-SWERL^®^	TRAKER™
	Winter	Spring/ Summer	Fall		Wind-Suspendable Dust Samples	Unpaved Road Dust Samples
Lemoore NAS	1/17/17 and 1/18/17 56 bulk soil samples	5/31/17 31 bulk soil samples	9/15/2017 29 bulk soil samples	5/31/17 (4 cores), 9/15/17 (3 cores)	9/15/17 (unpaved areas and roads near sites 1 [N26 and salt pit] and J [including dirt road west of site J]) (13 samples)	9/15/17 (unpaved roads around sites Lemoore [21 Str. north of Jackson Ave.], site 1 [N26], and 21 Str. to Bakersfield) (3 samples)
Edwards AFB	2/2/17 36 bulk soil samples	5/20/17 39 bulk soil samples	9/12/17 19 bulk soil samples	5/20/17 (5 cores), 9/12/17 (2 cores)	9/12/17 (unpaved roads near sites 5, 7, and 8) (3 samples)	9/12/17 (unpaved roads around sites 5 and 7) (3 samples)
29 Palms MCAGCC	1/28/17 43 bulk soil samples	6/10/17 and 6/11/17 60 bulk soil samples	10/13/17 and 10/14/17 76 bulk soil samples	6/10/17 and 6/11/17 (7 cores), 10/13/17 and 10/14/17 (6 cores)	10/13/17 (sites 5–10, and UT) (18 samples)	10/13/17 (unpaved roads between sites 7 and 10) (1 sample)

**Table 2 ijerph-17-05285-t002:** Results of *Coccidioides* DNA testing at sampling sites near NAS Lemoore.

Location	Soil Type and Soil Map Unit	Site Description	Coordinates		*Coccidioides* DNA *
LM-1 (N26)	Panoche clay loam, saline-alkali (151)	dirt road between agricultural fields and eucalyptus forest towards a dry, salty lakebed	36°22′38.96″ N	119°55′56.82″ W	pos
LM-2	Vanguard sandy loam, partially drained (168)	agricultural field	36°19′42.28″ N	119°52′06.31″ W	neg
LM-3	water, either Vanguard sandy loam, partially drained (168), or Gepford clay, partially drained (115)	area with ponds near creek, salty soils, plenty of Salt Bush (*Atriplex* spp.), high animal activity (birds and small mammals)	36°22′46.92″ N	119°55′17.69″ W	pos
LM-4	Gepford clay, partially drained (115)	dirt road along small canal, adjacent to new orchard	36°21′17.57″ N	119°54′18.29″ W	pos
LM-5	Grangeville fine sandy loam, saline-alkali, partially drained (121)	orchard with young almond trees	36°21′56.02″ N	119°52′19.85″ W	neg
LM-482	Calflax clay loam, saline-sodic, wet, 0–1% slope (482)	agricultural field	36°20′33.22″ N	120°02′55.25″ W	neg
LM-475	Posochaet clay loam, saline-sodic, wet 0–2% slopes (479)	agricultural field	36°19′41.27″ N	120°02′29.11″ W	neg
LM-479-1	Cerini clay loam, 0–2% slope (479)	agricultural field	36°15′20.05″ N	120°00′59.29″ W	neg
LM-479-2	Cerini clay loam, 0–2% slope (479)	agricultural field	36°17′24.04″ N	120°06′11.88″ W	neg
LM-462	Ciervo, wet Ciervo complex, saline-sodic, 0–1% slope (462)	meadow	36°16′11.35″ N	120°06′11.95″ W	neg
LM-477	Westhaven clay loam, 0–2% slope (477)	almond orchard	36°15′18.65″ N	120°05′03.48″ W	neg
LM-435	Lethenet clay loam, 0–2% slope (139)	agricultural field	36°15′19.62″ N	119°58′56.71″ W	neg
LM139-1	Lethenet clay loam (139)	unused agricultural field, grasses, clay rich area	36°14′04.20″ N	119°53′13.60″ W	neg
LM139-2	Lethenet clay loam (139)	abandoned agricultural field, high grasses and shrubs	36°15′20.48″ N	119°55′23.09″ W	neg
LM-115	Gepford clay, partially drained (115)	unused agricultural fields with grass	36°15′26.17″ N	119°55′23.09″ W	neg
LM-151	Calflax clay loam, saline-sodic, 0 to 2% slopes (151)	agricultural field	36°11′53.63″ N	119°56′04.02″ W	neg
LM-119	Grangeville sandy loam, saline-alkali (119)	along a dirt road between agricultural fields towards a meadow	36°16′18.55″ N	119°50′08.02″ W	pos
LM-J	Lemoore sandy loam, partially drained (137) Boggs sandy loam, partially drained (103)	salty environment with Iodine Bush (*Allenrolfea occidentalis*), was flooded in winter, rodent activity was observed	36°15′16.63″ N	119°48′40.00″ W	pos

* Sites were indicated as positive for the pathogen’s DNA when at least one out of three soil samples tested positive via nested PCR (with at least one diagnostic primer pair).

**Table 3 ijerph-17-05285-t003:** Results of *Coccidioides* DNA testing at sampling sites west of Edwards AFB adjacent to Antelope Acres.

Location	Soil Type and Soil Map Unit	Site Description	Coordinates		Detection of *Coccidioides* DNA *
AA-1	Adelanto coarse sandy loam, 2 to 5% slopes (AcA)	grassland, mostly invasive *Bromus* spp. and native and non-native annuals	34°44′48.97″ N	118°18′57.87″ W	neg
AA-2	Cajon loamy sand, loamy substratum, 0 to 2% slopes (CbA)	grassland, mostly invasive *Bromus* spp. and native and non-native annuals	34°44′23.01″ N	118°18′59.69″ W	neg
AA-3	Cajon loamy sand, loamy substratum, 0 to 2% slopes (CbA)	grassland, mostly invasive *Bromus* spp. and native and non-native annuals	34°43′59.80″ N	118°18′59.03″ W	neg
AA-4	Cajon loamy sand, 0 to 2% slopes (CaA)	eroded landscape with scattered vegetation, mostly native and non-native annuals	34°44′04.22″ N	118°18′25.29″ W	neg
AA-5	Cajon loamy sand, loamy substratum, 0 to 2% slopes (CbA)	eroded landscape with scattered vegetation, mostly native and non-native annuals	34°44′20.40″ N	118°18′24.36″ W	pos
AA-6	Cajon loamy sand, 0 to 2% slopes (CaA)	eroded landscape with scattered vegetation, mostly native and non-native annuals	34°44′34.60″ N	118°18′24.82″ W	pos
AA-7	Adelanto coarse sandy loam, 2 to 5% slopes (AcA)	eroded area dominated by rabbit brush (*Ericameria nauseosa*)	34°44′42.86″ N	118°18′25.24″ W	pos
AA-8	Cajon loamy sand, loamy substratum, 0 to 2% slopes (CbA)	eroded area dominated by rabbit brush (*Ericameria nauseosa*)	34°44′50.91″ N	118°18′16.91″ W	neg
AA-9	Cajon loamy sand, loamy substratum, 0 to 2% slopes (CbA)	eroded landscape with scattered vegetation, mostly native and non-native annuals	34°44′48.56″ N	118°18′03.94″ W	neg
AA-10	Cajon loamy sand, loamy substratum, 0 to 2% slopes (CbA)	eroded landscape with scattered vegetation including tumble weeds (*Salsola* sp.)	34°44′35.76″ N	118°18′03.89″ W	neg
AA-11	Cajon loamy sand, loamy substratum, 0 to 2% slopes (CbA)	abandoned agricultural field with scattered growth of invasive grasses (*Bromus* spp. and others)	34°44′11.53″ N	118°17′55.99″ W	neg
AA-12	Cajon loamy sand, 0 to 2% slopes (CaA)	grassland, mostly invasive *Bromus* spp. and native and non-native annuals	34°43′58.90″ N	118°18′10.16″ W	neg

* Sites were indicated as positive for the pathogen’s DNA when at least one out of three soil samples tested positive via nested PCR (with at least one diagnostic primer pair).

**Table 4 ijerph-17-05285-t004:** Results of *Coccidioides* DNA testing at sampling sites near MCAGCC Twentynine Palms.

Location	Soil Type and Soil Map Unit	Site Description	Coordinates		Detection of *Coccidioides* DNA *
29P-1	Not available ^&&^	landscape with scattered Creosote (*Larrea tridentata*) and Salt Bush (*Atriplex* spp.)	34°22′08.46″ N	116°32′24.39″ W	pos
29P-2	Not available ^&&^	landscape with scattered Creosote (*Larrea tridentata)* and Salt Bush (*Atriplex* spp.)	34°21′40.37″ N	116°31′55.75″ W	pos
29P-3	Not available ^&&^	area with large boulders, graffiti, littered with trash	34°20′05.00″ N	116°29′18.79″ W	pos
29P-4	Not available ^&&^	landscape with scattered Creosote (*Larrea tridentata*) and Salt Bush (*Atriplex* spp.) including occasional Joshua trees *(Yucca brevifolia*)	34°18′37.24″ N	116°28′15.72″ W	pos
29P-5	Not available ^&&^	landscape with scattered Creosote (*Larrea tridentata*) and Salt Bush (*Atriplex* spp.) including occasional Joshua trees *(Yucca brevifolia*), very rocky	34°17′10.79″ N	116°27′13.62″ W	pos
29P-6	Not available ^&&^	landscape with scattered Creosote (*Larrea tridentata*) and Salt Bush (*Atriplex* spp.) including occasional Joshua trees *(Yucca brevifolia*), eroded soil	34°15′35.74″ N	116°26′23.01″ W	pos
29P-7	Not available ^&&^	landscape with scattered Creosote (*Larrea tridentata*) and Salt Bush (*Atriplex* spp.), lots of ants	34°08′11.17″ N	116°01′04.85″ W	pos
29P-8	Not available ^&&^	landscape with scattered Creosote (*Larrea tridentata*) and Salt Bush (*Atriplex* spp.), eroded soil, rodent activity high	34°08′11.85″ N	115°00′09.16″ W	pos
29P-9	Not available ^&&^	landscape with scattered Creosote (*Larrea tridentata*) and Salt Bush (*Atriplex* spp.), some Beavertail cacti (*Opuntia basilaris*)	34°08′13.90″ N	115°58′10.91″ W	pos
29P-10	Not available ^&&^	landscape with scattered Creosote (*Larrea tridentata*) and scattered Salt Bush (*Atriplex* spp.), lots of ants	34°08′14.75″ N	115°54′50.02″ W	pos
UT	Not available ^&&^	eroded landscape, scattered Salt Bush (*Atriplex* spp.), mostly unvegetated	34°11′41.84″ N	116°02′06.72″ W	pos
SR	Desertqueen-Jumborox-Rock outcrop association, 2 to 8% slopes, warm	Skull Rock trail, boulders, lots of organic matter, Desert Oaks (*Quercus* sp.) and many other shrubs	33°59′50.28″ N	116°03′37.44″ W	neg
HH	Morongo loamy sand, 2 to 4% slopes	Hall of Horrors Trail, scattered Joshua trees, Salt Bush (*Atriplex* spp.) and other shrubs	34°00′02.16″ N	116°08′45.96″ W	pos
HV	Rock outcrop	Hidden Valley, lots of organic matter, Desert oaks, shrubs and yucca plants	34°00′12.00″ N	116°10′25.32″ W	neg

* Sites were indicated as positive for DNA of the pathogen when at least one out of three soil samples tested positive via nested PCR (with at least one diagnostic primer pair). ^&&^ No digital data available in the USDA websoilsurvey database.

**Table 5 ijerph-17-05285-t005:** Diagnostic PCR results for all soil DNA extracts (*n =* 389) (ITSC1Af/r and EC3f/EC100r).

Detection of *Coccidioides* spp.	NAS Lemoore		Antelope Acres (West of EAFB)		Twentynine Palms (South of MCAGCC)	
	*n*	%	*n*	%	*n*	%
*n* (389)	116	100	94	100	179	100
positive (with at least one diagnostic primer pair)	32	27.59	14	14.89	76	42.46
positive (with both diagnostic primer pairs)	11	9.48	11	11.7	42	23.46
negative (with both diagnostic primer pairs)	73	62.93	69	73.41	61	34.08

**Table 6 ijerph-17-05285-t006:** Detection of *Coccidioides* DNA in wind-suspendable and vehicle-suspendable dust samples collected around NAS Lemoore, Antelope Acres (west of EAFB) and MCAGCC Twentynine Palms (*n =* 13 [PI-SWERL^®^], *n =* 6 [TRAKER™]).

Site	Sample Type	Soil Type and Map Unit	Coordinates	Detection of *Coccidioides* DNA.
**Lemoore**				
21. Street to north of Jackson Ave., Lemoore	TRAKER	Panoche clay loam, saline-alkali (151)		neg
site 1, N-26, along dirt road	TRAKER	Panoche clay loam, saline-alkali (151)	36°22′37.02″ W, 119°55′56.89″ W	neg
Site 1, N-26, dirt road	PI-SWERL	Panoche clay loam, saline-alkali (151)	36°22′37.02″ N, 119°55′56.89″ W	neg
Site 2, dry lake	PI-SWERL	Panoche clay loam, saline-alkali (151)	36°22′18.19″ N, 119°55′54.16″ W	neg
Site 3, near creek, saltbush area	PI-SWERL	Vanguard sandy loam/Gepford clay (partially drained), (168/115)	36°15′20.74” N, 119°50′04.34″ W	neg
Site J, iodine bush area	PI-SWERL	Lemoore sandy loam/Boggs sandy loam (partially drained), (137/103)	36°15′16.81″ N, 119°48′35.82″ W	pos
21. street, Lemoore to Bakersfield, near intersection to Hwy. 58	TRAKER			neg
**Antelope Acres**				
Site 7	PI-SWERL	Adelanto coarse sandy loam (CaA)	34°44′48.12″ N, 118°18′24.80″ W	neg
Site 7	TRAKER	Adelanto coarse sandy loam (CaA)	34°44′48.12″ N, 118°18′24.80 W	neg
Site 5	PI-SWERL	Cajon loamy sand (CaA)	34°44′21.34″ N, 118°18′13.64″ W	pos
dust on car after sampling (site 8)	swab		34°44′50.93″ N, 118°18′16.92″ W	neg
**Twentynine Palms**				
Site 5	PI-SWERL	no data	34°15′56.66″ N, 116°26′45.20″ W	pos
Site 6	PI-SWERL	no data	34°15′33.01″ N, 116°26′26.41″ W	neg
Site 7	PI-SWERL	no data	34°08′12.34″ N, 116°01′0959″ W	neg
Site 8	PI-SWERL	no data	34°08′08.99″ W, 116°00′10.62″ W	pos
Site 9	PI-SWERL	no data	34°08′09.85″ N, 115°58′18.46″ W	pos
Site 10	PI-SWERL	no data	34°08′03.59″ N, 115°54′41.76″ W	pos
**Dirt roads between sites 7 and 10**	TRAKER	no data	see coordinates above	pos
Site UT	PI-SWERL	no data	34°11′44.84″ N, 116°02′11.04″ W	neg
Site UT	TRAKER	no data	34°11′44.84″ N, 116°02′11.04″ W	neg

**Table 7 ijerph-17-05285-t007:** Positive diagnostic PCR results (ITSC1f/r and EC3f/EC100r) for all DNA extracts from soil core samples (cores: *n =* 25, individual soil samples: *n =* 146) collected around NAS Lemoore, Antelope Acres and MCAGCC Twentynine Palms in percent.

		Soil Core Samples (a–f) and Depth (cm)											
Sites and Number of Cores	*n*	a 0–2	b 5–7	c 10–12	d 18–20	e 23–25	f 28–30	a 0–2	b 5–7	c 10–12	d 18–20	e 23–25	f 28–30
		Indicated positive by one diagnostic primer pair (%)						Indicated positive with both diagnostic primer pairs in agreement (%)					
Lemoore, 5 cores	26	11.5	11.5	7.7	7.7	0	3.9	7.7	3.9	3.9	3.9	0	0
Antelope Acres, 7 cores	40	0	0	2.5	0	0	0	0	0	2.5	0	0	0
Twentynine Palms, 13 cores	80	11.3	6.3	5.0	5.0	1.3	2.5	5	2.5	2.5	2.5	1.3	1.3

**Table 8 ijerph-17-05285-t008:** Summary of soil analyses for soil chemical and physical parameters and detection of *Coccidioides* DNA in soil and dust samples. A brief site description with indication of dominant vegetation and risk of exposure to the pathogen is indicated, as well.

Locations	Soil Analyses (Parameter Averages or Ranges)		Detection of *Coccidioides* DNA (Both Diagnostic Primer Pairs Agreed)		Environment/Habitat		Exposure Risk
Site Description	Soil Chemical Analyses	Soil Physical Analyses	Soil	Dust	Landform	Vegetation	Presumed Risk of *Coccidioides* Exposure *
**Lemoore:** dominated by agricultural activities, ~6.4 in rainfall, soil types: clay loam, sandy loam, fine sandy loam, clay, many saline-sodic and alkaline	pH: 7.8–8.1, EC (mS/m^3^): 1 (winter)–13 (summer), TDS (g/L): 0.4–6.1 (winter–summer), CaCO_3_ (%): 3.52, high levels of SO_4_^2−^	sand (39.7%), silt (53.7%), clay (6.6%)	9.48%	site 4 (PI-SWERL)	alluvial fans	mostly agricultural fields and orchards, non-native trees, grasses, meadows, few semi-natural areas with Salt Bush and Iodine Bush	low
**Antelope Acres (west of EAFB):** western Mojave Desert, declining agricultural activities, former ranchland, increase in soil disturbance due to renewable energy construction, eroded soils, windy, ~7.4 in rainfall, soil types: coarse sandy loam, loamy sand	pH: 7.1–7.2, EC (mS/m^3^): <1 (winter–summer), TDS (g/L): <0.1 (winter–summer), CaCO_3_ (%): 0.05, high levels of K	sand (63%), silt (32.9%), clay (4.1%)	11.70%	site 5 (PI-SWERL)	alluvial fans, dry lakes	invasive grasses, rabbit brush, Salt Bush, native and non-native herbs and wildflowers, eroded former farmland and non-vegetated areas (solar ranches)	high, increasing
**Twentynine Palms:** Southern Mojave Desert, semi-disturbed desert, windy, no agriculture, scattered settlements, ~4.4 in rainfall, outcrop associations, loamy sand, coarse loamy sand	pH: 7.5–8, EC (mS/m3): <1 (winter–summer), CaCO_3_ (%): 0.65, high levels of K and Ca	**29 Palms:** sand (66.1%), silt (30.5%), clay (3.4%)	23.46%	sites 5, 8, 10 (PI-SWERL), sites 7–10 TRAKER	fan aprons, alluvial fans	desert shrub, Creosote, Salt Bush, Joshua Trees, cacti, Desert Oak and pines, few invasive grasses, native and non-native herbs including wildflowers	high

* Prevalence of positive sites based on samples analyzed.

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
