# Peer review of "Valley Fever: Environmental Risk Factors and Exposure Pathways Deduced from Field Measurements in California"

_ijerph, 2020, doi:10.3390/ijerph17155285_

Round 1

Reviewer 1 Report

General comments: manuscript is very diffuse and unfocused. The main issue is the reported goal of the paper was to detect Coccidioides on military bases, but then there are several places where the data/results are generalized- and fairly broad statements are made about the environmental parameters that influence the growth/presence of the fungus

Methods section reads more like a results section in many places. The first paragraph of this section is confusing, and should be deleted (I think most of this is in the table, or is unnecessary. 2.1 first 2 paragraphs again seem like results not methods, and most seems unnecessary for the methods section. Section 2.3 - there is a discussion of sensitivity and specificity that seems editorial/discussion and there is no support presented for statements. that last paragraph should be deleted unless there is some evidence to support. And it is irrelevant for the manuscript, you are using the assay you developed, no justification needed. 2.4- citation for "established techniques" or summarize what you mean here. What is "degree of disturbance" and how exactly is that measured?

Will the authors submit the sequence files to NCBI or make that data available beyond 10 samples? I'd like to see the alignments or some indication of the similarity. Are the authors concerned that the sequences match C. posadasii and not C. immitis?

Throughout the authors say that they found the organism in in the soil- but be careful to clearly state the DNA was detected, and whether or not the organism is actively growing is not known. (dormant spores vs. mycelia)

Table 5- do you mean positive with ONLY one PCR (second line)

line 406- is "quiet" quite here?

The statistical test applied (students t-test) does not appear to be correct- you are testing multiple variables and they should be analyzed together. Likely many of these are correlated with each other? 

The analysis of environmental data seem generalized beyond the regions currently sampled. The design of the study was not set up to answer a specific hypothesis and is rather descriptive in nature. I suggest the authors revise the manuscript to be focused on reporting on their observations and avoid trying to fit a model (disturbed vs. not) and risk assessments to their limited data (Fig 13). If more robust analysis was conducted, these details are not clear from the methods section.

Author Response

Dear Reviewer 1, please see our response to all your comments and suggestions in the attached pdf file. Thank you for your feedback!

Reviewer 2 Report

The article “Valley fever: Environmental risk factors and exposure pathways deduced from field measurements in California” of Antje Lauer, et al., code: ijerph-812506, try to explain in detail the potential relationship between coccidioidomycosis and soil and dust near three military bases in California. The aim of the paper is interesting and very important to the soldiers in the three military bases and nearby residents, even similar situations elsewhere around the world. So the manuscript can be published in this journal after minor corrections which are:

  1. Most of the figures are not clear, so improve or re-worked the figures (such as fig.1 to fig. 10…) to make sure they are clearer in the revised manuscript.
  2. As a practice in scientific papers, there should be no figures in conclusions.
  3. About the differences of coccidioidomycosis among the three main sampling regions, how do you rule out that it's not caused by climate?

Author Response

Dear Reviewer 2, thank you for your comments. Please have a look at the attached pdf. We hare responded to all your points. Thank you!

Reviewer 3 Report

This manuscript is well-written. The whole study is meaningful in identifying the environmental risks of valley fever. I do not have many comments. My overall suggestion is to make the Method section concise and straightforward. The Method section is too redundant, which is difficult for general environmental researchers to understand the entire sampling and lab process.

Author Response

Dear Reviewer 3, please find our response to your comments and suggestions in the attached pdf. Thank you!
